# Magnetic Resonance Imaging and Spectroscopy Analysis in a Pelizaeus–Merzbacher Disease Rat Model

**DOI:** 10.3390/diagnostics12081864

**Published:** 2022-08-02

**Authors:** Maho Ishikawa, Reika Sawaya, Miki Hirayama, Junpei Ueda, Shigeyoshi Saito

**Affiliations:** 1Department of Medical Physics and Engineering, Area of Medical Imaging Technology and Science, Division of Health Sciences, Osaka University Graduate School of Medicine, Osaka 565-0871, Japan; maho.i@sahs.med.osaka-u.ac.jp (M.I.); u010443b@ecs.osaka-u.ac.jp (R.S.); uedaj@sahs.med.osaka-u.ac.jp (J.U.); 2Department of Medical Technology, Division of Radiology, Osaka University Hospital, Osaka 565-0871, Japan; m_hirayama@hp-rad.med.osaka-u.ac.jp; 3Department of Advanced Medical Technologies, National Cerebral and Cardiovascular Center Research Institute, Osaka 564-8565, Japan

**Keywords:** Pelizaeus–Merzbacher disease, dysmyelination, magnetic resonance imaging, ^1^H-magnetic resonance spectroscopy

## Abstract

Pelizaeus–Merzbacher disease (PMD) is an X-linked recessive disorder of the central nervous system. We performed 7 Tesla magnetic resonance imaging of the brain in Tama rats, a rodent PMD model, and control rats, as well as evaluated the diagnostic values. In the white matter of the Tama rats, the T_2_ values were prolonged, which is similar to that observed in patients with PMD (60.7 ± 1.8 ms vs. 51.6 ± 1.3 ms, *p* < 0.0001). The apparent diffusion coefficient values in the white matter of the Tama rats were higher than those of the control rats (0.68 ± 0.03 × 10^−3^ mm^2^/s vs. 0.64 ± 0.03 × 10^−3^ mm^2^/s, *p* < 0.05). In proton magnetic resonance spectroscopy, the N-acetylaspartate (6.97 ± 0.12 mM vs. 5.98 ± 0.25 mM, *p* < 0.01) and N-acetylaspartate + N-acetylaspartylglutamate values of the Tama rats were higher (8.22 ± 0.17 mM vs. 7.14 ± 0.35 mM, *p* < 0.01) than those of the control rats. The glycerophosphocholine + phosphocholine values of the Tama rats were lower than those of the control rats (1.04 ± 0.09 mM vs. 1.45 ± 0.04 mM, *p* < 0.001). By using Luxol fast blue staining, we confirmed dysmyelination in the Tama rats. These results are similar to those of patients with PMD and other PMD animal models.

## 1. Introduction

Myelin in the central nervous system (CNS) plays an important role in neural survival [1]. Pelizaeus–Merzbacher disease (PMD) is an X-linked recessive disorder affecting myelin formation in the CNS and is caused by mutations in the proteolipid protein 1 gene [2]. Male patients with PMD generally show severe dysmyelination in the CNS, but women patients carrying mutated genes are asymptomatic [3,4]. Several animal models for PMD, including jimpy [5] and myelin synthesis-deficient (MSD) mice, are available [6]. A neurological mutation in Wistar rats transmitted by an X-linked recessive lethal gene leads to myelin-deficient (MD) rats, i.e., Tama rats [7,8]. The first symptom includes head tremor, which occurs at 12–15 days of age. Tremors become systemic within a few days and disappear when the animals are at rest. From days 17 to 21, the slightest disturbance induces a generalized seizure. Finally, the MD rats die within 30 days after birth. The color of the spinal cord of normal rats is white, whereas that of MD rats is gray. This is the only difference that can be observed between the two rodent groups. By using microscopy, a total lack of myelin formation can be noted at all CNS levels [9].

Magnetic resonance imaging (MRI) assists in the diagnosis of PMD. Patients with PMD show T_2_ prolongation due to dysmyelination [10]. Bar-Shir et al. used MD rats and concluded that the lack of myelin affects high-b-value q-space diffusion imaging and conventional diffusion tensor imaging [11]. Takanashi et al. used MSD mice to perform proton magnetic resonance spectroscopy (^1^H-MRS) [12]. They found that the concentrations of N-acetylaspartate (NAA) and N-acetylaspartylglutamate (NAAG) increased, whereas that of choline (Cho) decreased. They concluded that ^1^H-MRS may be an important marker of PMD. Myelin has a multilayer membrane structure, and bound water molecules are present between the phospholipid layers in myelin [13,14,15,16]. We assumed that dysmyelination could decrease the ratio of bound water molecules and increase the ratio of free water molecules. Apparent diffusion coefficient (ADC) mapping shows the state of motion of water molecules [17,18]. We expected that the ADC values of myelin would increase. However, few studies on MD rats have been conducted, and the relationship between ADC values and dysmyelination has not been revealed.

In this study, we used another type of MD rat, known as Tama. We aimed to evaluate the findings of T_2_-weighted imaging (T_2_WI), ADC mapping, ^1^H-MRS, and Luxol fast blue (LFB) staining between the Tama and control rats.

## 2. Materials & Methods

### 2.1. Animals

All experimental protocols were approved by the Research Ethics Committee of Osaka University (R02-05-0). All experimental procedures involving animals and their care were performed in accordance with the Osaka University Guidelines for the Care and Use of Laboratory Animals. Four 3-week-old Tama rats with spontaneous mutations in the genes (weight = 54.8 ± 0.4 g) and five age-matched control rats (weight = 67.6 ± 4.3 g) were used. All the rats were housed in a controlled vivarium environment (24 °C; 12:12-h light:dark cycle) with free access to food and water.

### 2.2. MRI Measurements

For these experiments, we used a horizontal 7.0 Tesla MRI (PharmaScan 70/16 US; Bruker Biospin, Ettlingen, Germany) equipped with an inner diameter of 30-mm volume coil. MRI sequences and ^1^H-MRS were performed for all rats [19]. During MRI scanning, all rats were anesthetized with isoflurane (3.0% for induction and 2.0% for maintenance). They were positioned in a stereotaxic frame, and their body temperature was continuously monitored using a physiological monitoring system (SA Instruments Inc., Stony Brook, NY, USA), maintained at 36.5 °C with regulated water flow.

Axial T_2_-mapping images were initially acquired (multislice spin echo; repetition time [TR]/echo time [TE] = 2250/8.3 ms, 12 echoes; slice thickness = 0.5 mm; matrix = 256 × 256; field of view = 36 × 36 mm; resolution = 141 µm; slices = 20; scan time = 9 min and 36 s). Diffusion-weighted imaging was performed (spin echo; TR/TE = 2000/26.5 ms; slice thickness = 0.5 mm; matrix = 128 × 128; field of view = 36 × 36 mm; resolution = 282 µm; slices = 20; b-value = 1000, 2000 s/mm^2^; segment = 4; axis = 30; duration = 5 ms; scan time = 8 min and 40 s) [20,21].

Images of the same resolution were then used to accurately position a voxel of 3 × 3 × 3 mm^3^ in both the hippocampi. Magnetic field homogeneity was ascertained using the Bruker Mapshim shimming protocol, and good shimming between 8.9 and 12.1 Hz was achieved in the voxels. ^1^H-MRS was performed using a point-resolved spectroscopy sequence (TR/TE = 2500/20 ms) combined with variable power and optimized relaxation delays (VAPOR) water suppression. Metabolite spectra were acquired using 256 repetitions with VAPOR and 32 repetitions without VAPOR for a total scan time of 12 min. The metabolite concentrations of NAA + NAAG, creatine + phosphocreatine (Cr + PCr), lactate (Lac), Cho, glutamate (Glu), glutamine (Gln), and myoinositol (Ins) were quantified using the basic setting of the LC model [22].

We set regions of interest (ROIs) in the axial T_2_WI from an anatomic point of view [23], measuring the T_2_ values of the ROIs in T_2_ mapping with ImageJ software, an analysis tool [24]. The ROIs included the cerebral cortex, white matter, striatum, corpus callosum, and hippocampus. Figure 1 shows an example of a T_2_ image. Similarly, we set the ROIs and measured the ADC values on the ADC maps.

### 2.3. LFB Staining

We removed the brains of two Tama rats and two control rats after MRI measurements and perfused them with 4% phosphate-buffered saline. Axial slices (5–10-µm thick) were stained with LFB.

### 2.4. Statistical Analysis

Data are presented as mean ± standard deviation. Differences between Tama and control rats were compared using an unpaired t-test. Statistical significance was set at *p* < 0.05. All statistical analyses were performed using Prism, version 9 (GraphPad Software, San Diego, CA, USA).

## 3. Results

### 3.1. MRI Findings, T_2_ Value Measurements, and ADC Value Measurements

Axial T_2_-weighted images, T_2_ maps, and ADC maps of the Tama and control rats are shown in Figure 2. The signal intensity in the white matter of the Tama rats (Figure 2A) was lower than that of the control rats (Figure 2D). The average T_2_ values are shown in Figure 3. The average T_2_ values of the Tama rats in the cerebral cortex (54.9 ± 1.3 ms vs. 52.6 ± 0.8 ms, *p* < 0.05), white matter (60.7 ± 1.8 ms vs. 51.6 ± 1.3 ms, *p* < 0.0001), striatum (55.9 ± 1.3 ms vs. 52.7 ± 0.9 ms, *p* < 0.001), and corpus callosum (65.7 ± 4.7 ms vs. 56.8 ± 2.8 ms, *p* < 0.001) were longer than those of the control rats and showed significant differences. However, the average T_2_ values in the hippocampus showed no significant difference (50.7 ± 1.7 ms vs. 50.6 ± 1.1 ms).

The average ADC values are shown in Figure 4. The average ADC values in the white matter of the Tama rats were larger than those of the control rats and showed a significant difference (0.68 ± 0.03 × 10^−3^ mm^2^/s vs. 0.64 ± 0.03 × 10^−3^ mm^2^/s, *p* < 0.05). The average ADC values in the cerebral cortex (0.61 ± 0.04 × 10^−3^ mm^2^/s vs. 0.63 ± 0.01 × 10^−3^ mm^2^/s), striatum (0.59 ± 0.03 × 10^−3^ mm^2^/s vs. 0.61 ± 0.01 × 10^−3^ mm^2^/s), corpus callosum (0.72 ± 0.04 × 10^−3^ mm^2^/s vs. 0.71 ± 0.03 × 10^−3^ mm^2^/s), and hippocampus (0.61 ± 0.03 × 10^−3^ mm^2^/s vs. 0.60 ± 0.04 × 10^−3^ mm^2^/s) showed no significant differences between the Tama and control rats.

### 3.2. Brain Metabolites Measured Using ^1^H-MRS

The concentrations and %SD of each metabolite in the Tama and control rats are shown in Figure 5. The values of NAA (6.97 ± 0.12 mM vs. 5.98 ± 0.25 mM, *p* < 0.01) and NAA + NAAG (8.22 ± 0.17 mM vs. 7.14 ± 0.35 mM, *p* < 0.01) of the Tama rats were higher than those of the control rats and showed significant differences. The GPC + PCh values of the Tama rats were lower than those of the control rats and showed a significant difference (1.04 ± 0.09 mM vs. 1.45 ± 0.04 mM, *p* < 0.001). The values of Ins (4.60 ± 0.78 mM vs. 5.39 ± 0.35 mM), NAAG (1.25 ± 0.05 mM vs. 1.16 ± 0.14 mM), Cr + PCr (5.91 ± 0.16 mM vs. 6.21 ± 0.31 mM), and Glu + Gln (12.14 ± 0.29 mM vs. 11.49 ± 0.89 mM) showed no significant differences between the Tama and control rats.

### 3.3. LFB Staining

The results of LFB staining for the Tama and control rat brains are shown in Figure 6. The Tama rat brain (A) showed no myelin staining, and the control rat brain (B) showed dense myelin staining in the white matter.

## 4. Discussion

In this study, we evaluated the findings of T_2_WI, ADC mappings, ^1^H-MRS, and LFB staining between the Tama and control rats. The main results were as follows: they showed dysmyelination, which caused the T_2_ values in some brain sites and the ADC values of the white matter to increase. Furthermore, the concentrations of NAA and NAA + NAAG increased, whereas the concentration of Cho decreased. These results are similar to those of patients with PMD and other PMD animal models.

The process of myelination depends on the species involved. In contrast to humans, myelination begins after birth in rats [25,26]. The rate of myelination varies across different brain sites. Myelination begins in the primary motor and sensory areas of the cerebral cortex and the corpus callosum, and then it spreads outward. At 24 days of age, almost all brain sites are myelinated, but myelination is not fully complete in the posterior tracts of the fornix and mammillothalamic tract [27]. In this study, we performed whole-brain MRI and measured the T_2_ values of some brain sites, as shown in Figure 1. We found that the T_2_ values of the white matter were prolonged, which confirms the findings of previous studies in MSD mice and humans [12,28]. As for the other sites, Junichi et al. showed that the T_2_ values of the cerebral cortex and striatum were prolonged in MSD mice [12]. We obtained the same results and found that the T_2_ value of the corpus callosum was prolonged, but that of the hippocampus did not show a significant difference. In response to the small size of the hippocampus and the difficulty in accurate segmentation in our study, the ROIs of the hippocampus might have included other fields.

The ADC value of the white matter showed a significant difference, but the other sites did not show significant differences. Myelin has multilayer membranes, and bound water molecules exist between phospholipid layers. The bound water molecules have short T_2_ relaxation times. Meanwhile, free water molecules are found in the intra- and extra-axonal spaces, which have long T_2_ relaxation times [29]. We believe that dysmyelination decreased the ratio of bound water, which caused T_2_ time prolongation and increased ADC values. The ADC values of the PMD model rats have not been reported. In a case report of PMD by Spencer, the ADC value of the white matter was higher than that of healthy people, but the ADC value of the cortex was normal [30]. Similar results were obtained in the present study. Multiple sclerosis (MS) is a common cause of dysmyelination. Patients with MS show higher ADC values in the white matter than healthy individuals [31,32]. By contrast, Degaonkar et al. used MS model rats and monitored the changes in the ADC values in their brain internal capsule areas from the early phase to the remyelination stage. They found that the ADC values in the focal demyelinating lesion were higher than those in the surrounding edematous area during the early phase of demyelination, but the ADC values in both areas decreased during remyelination [33]. Therefore, ADC values may be useful for diagnosing the degree of dysmyelination and distinguishing PMD from other disease models.

In ^1^H-MRS, Takanashi et al. used two different mouse models of hypomyelination, namely, MSD and shiverer mice [34]. They reported that the MRS results in the thalamus of the two types of patients were different. The concentrations of NAA + NAAG, Cr, Glu, and Gln were elevated, but the concentration of Cho was decreased in MSD mice compared with wild-type mice. In shiverer mice, the concentration of Cho decreased compared with that in wild-type mice, but the degree of Cho reduction was much lower than that in MSD mice. They explained that the difference in NAA + NAAG depends on the oligodendrocyte patterns. In MSD mice, oligodendrocyte apoptosis occurs before myelination begins [35]. This increases the number of oligodendrocyte progenitor cells while decreasing the number of mature oligodendrocytes. In shiverer mice, the absence of myelin basic proteins causes oligodendrocytes to fail to form a compact myelin sheath. Tama rats may have allelic genes such as those of jimpy and MSD mice. The result that the concentration of NAA + NAAG in the Tama rats increased was reliable in our study. Furthermore, they found that the concentration of NAAG increased in only a few MSD mice [12]. The separate measurement of NAA and NAAG by using MRS is often difficult because of the large superposition of both spectra, and its results are difficult to interpret. The concentration of NAAG was not significantly different in the present study. Takanashi et al. chose an ROI in the thalamus. Hhowever, we set an ROI in the hippocampus. Thus, the location of the ROI may have affected the results. In humans, metabolite concentrations are not necessarily consistent and vary depending on the form of the PMD. For example, patients with duplications have higher NAA concentration, whereas those with other mutations have lower NAA concentration [36]. In humans, a study by Hanefeld et al. also reported that the concentrations of NAA + NAAG, Gln, Ins, and Cr+ PCr in the affected white matter increase [37]. This suggests that the concentration of Cho decreases regardless of the state of PMD in humans, mice, and rats.

This study has two main limitations. First, MRI sequences were performed at only one time point. Monitoring the results of MRI sequences continuously may improve our current results and reveal the process of dysmyelination in more detail. Second, an optical microscope was used in the histochemical analysis, but an electronic microscope can reveal more about the relationship between our current results and the PMD.

In conclusion, we used Tama rats, a rat model of PMD, and revealed that the ADC values in the white matter can show the state of PMD. Moreover, the animal model of PMD with a deficiency of proteolipid protein showed increased concentrations of NAA and NAA + NAAG and decreased concentration of Cho, which is similar to patients with PMD.

## Figures and Tables

**Figure 1 diagnostics-12-01864-f001:**
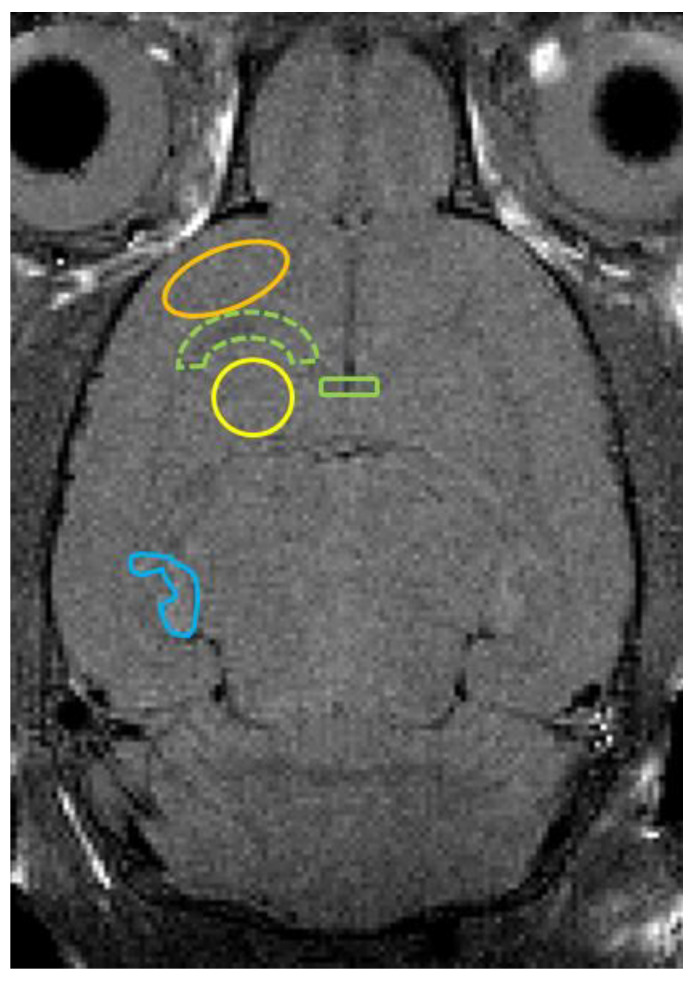
Axial T_2_-weighted image shows ROI placements on the cerebral cortex (orange line), white matter (green dotted line), striatum (yellow line), corpus callosum (green line), and hippocampus (blue line).

**Figure 2 diagnostics-12-01864-f002:**
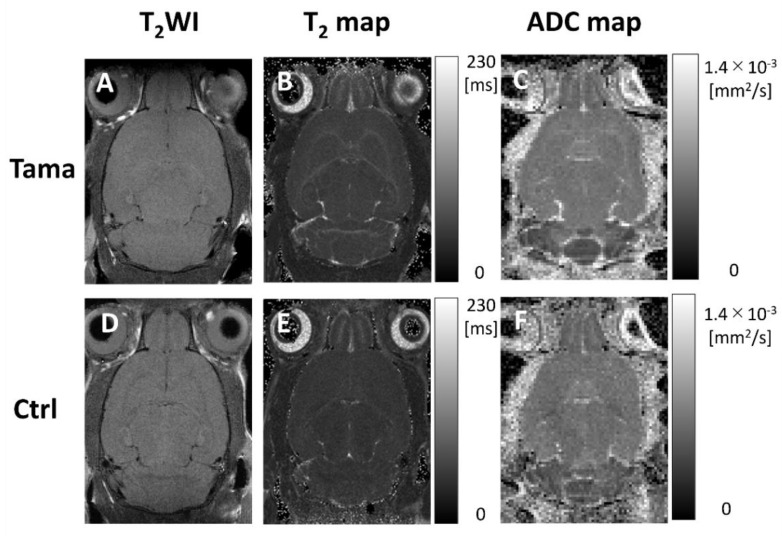
T_2_-weighted images, T_2_ maps, and ADC maps of the Tama rats and control rats (Ctrl). The white matter intensity of the Tama rat in the T_2_-weighted image (**A**) was higher than that of the control rat (**D**). In the T_2_ maps, the T_2_ values in the white matter of the Tama rat (**B**) were higher than those of the control rat (**E**). In the ADC maps, the ADC values in the white matter of the Tama rat (**C**) were higher than those of the control rat (**F**).

**Figure 3 diagnostics-12-01864-f003:**
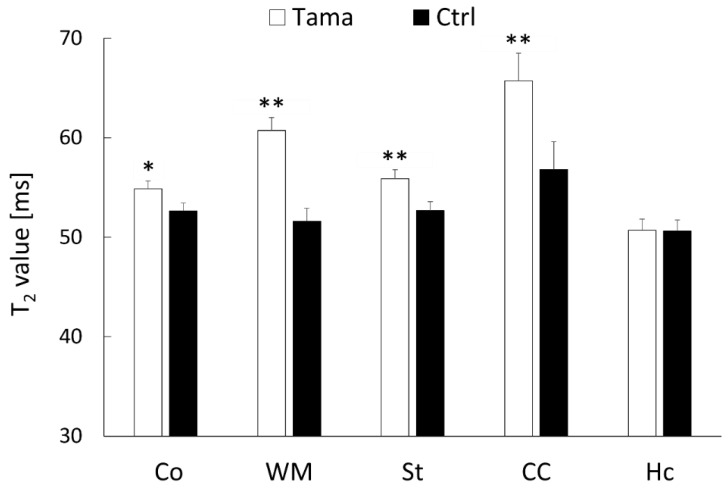
T_2_ values in the cerebral cortex (Co), white matter (WM), striatum (St), corpus callosum (CC), and hippocampus (Hc) were measured. The average values of the Tama and control rats (Ctrl) were measured. The values of the Tama rats showed * *p* < 0.05 and ** *p* < 0.01 compared with the control rats.

**Figure 4 diagnostics-12-01864-f004:**
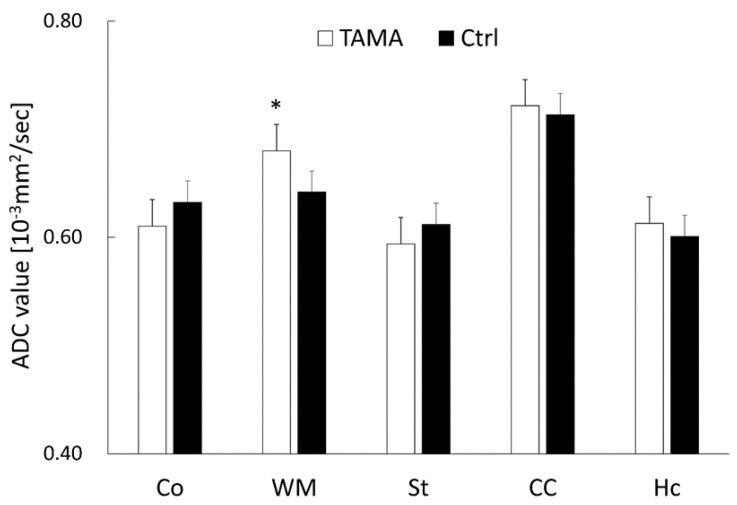
The ADC values in the cerebral cortex (Co), white matter (WM), striatum (St), corpus callosum (CC), and hippocampus (Hc). The average values of the Tama and control rats (Ctrl) were calculated. The values of the Tama rats showed * *p* < 0.05 compared with the control rats.

**Figure 5 diagnostics-12-01864-f005:**
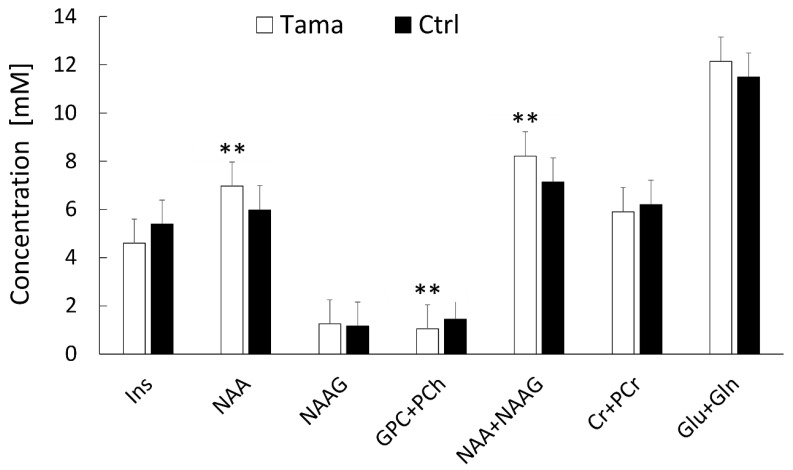
^1^H-MRS was performed, and the metabolite concentrations of the Tama and control rats (Ctrl) were measured. The metabolites included myoinositol (Ins), N-acetylaspartate (NAA), N-acetylaspartylglutamate (NAAG), glycerophosphocholine + phosphocholine (GPC +PCh), creatine + phosphocreatine (Cr + PCr), glutamate (Glu), and glutamine (Gln). The concentrations of NAA, GPC + PCh, and NAA + NAAG of the Tama rats showed ** *p* < 0.01 compared with the control rats.

**Figure 6 diagnostics-12-01864-f006:**
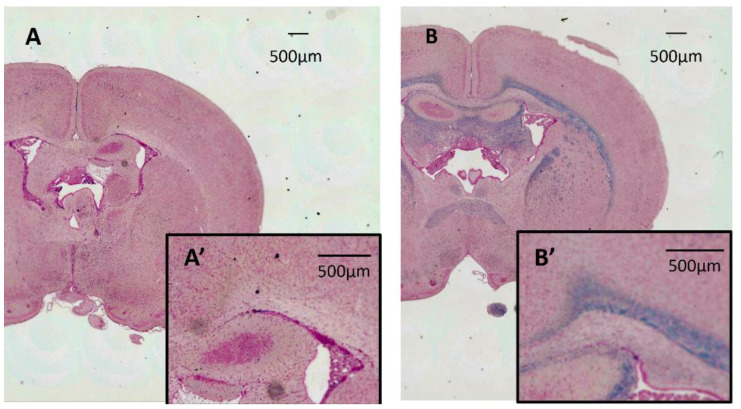
Results of Luxol fast blue staining for the Tama and control rat brains. The Tama rat brain (**A**) shows no myelin staining, and the control rat brain (**B**) shows dense myelin staining in the white matter. (**A′**,**B′**) are magnified myelin. Each of the scale bars shows 500 µm.

## Data Availability

The data presented in this study are available on request from the corresponding author.

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
