# Peer review of "Magnetic Resonance Imaging and Spectroscopy Analysis in a Pelizaeus–Merzbacher Disease Rat Model"

_diagnostics, 2022, doi:10.3390/diagnostics12081864_

Round 1

Reviewer 1 Report

These authors have used the magnetic resonance imaging to examine the brain in Tama rats, an Pelizaeus–Merzbacher disease animal model. They have conducted imaging and data analysis of different regions within the brain. Overall, their experimental design and data are solid. However, there are several issues needed to be further clarified before publication in the journal. 

Major:

1. The data regarding the apparent diffusion coefficient (ADC) mapping are intriguing. In Fig. 4, the author claimed that the ADC value of white matter in Tama mice was larger than the control. However, given the similar variances among different regions of Tama and control mice, I cannot see why there were statistically different in WM, and why there were no difference in other brain regions.  This is important since the author concluded that the ADC  values in the white matter can show the state of PMD.

2. Similarly, I also have doubt regarding the shown differences in NAA, GPC+PCH, NAA+NAAG, but not in Ins, Glu+Gln.  

Author Response

Comments and Suggestions for Authors

These authors have used the magnetic resonance imaging to examine the brain in Tama rats, an Pelizaeus–Merzbacher disease animal model. They have conducted imaging and data analysis of different regions within the brain. Overall, their experimental design and data are solid. However, there are several issues needed to be further clarified before publication in the journal.

Thank you for your considerate suggestions for and comments on our manuscript. We have revised the manuscript based on your suggestions. The following paragraph has been added to the manuscript.

Major:

  1. The data regarding the apparent diffusion coefficient (ADC) mapping are intriguing. In Fig. 4, the author claimed that the ADC value of white matter in Tama mice was larger than the control. However, given the similar variances among different regions of Tama and control mice, I cannot see why there were statistically different in WM, and why there were no difference in other brain regions. This is important since the author concluded that the ADC values in the white matter can show the state of PMD.

Thank you for your considerate suggestions for and comments on our manuscript. Fig. 4, we claimed that the ADC value of white matter in Tama mice was larger than the control. As you mention, there are the similar variances among different regions of Tama and control mice in Fig. 4. Our description of the SD in Fig.4 was incorrect. So, We have revised Fig.4 and Fig.4 description.

Line 137,

The average ADC values in the white matter of the Tama rats were larger than those of the control rats and showed significant difference (0.68 ± 0.04 × 10-3 mm2/s vs. 0.64 ± 0.03 × 10-3 mm2/s, p < 0.05). While, the average ADC values in the cerebral cortex (0.61 ± 0.03 × 10-3 mm2/s vs. 0.63 ± 0.01 × 10-3 mm2/s), striatum (0.59 ± 0.03 × 10-3 mm2/s vs. 0.61 ± 0.01 × 10-3 mm2/s), corpus callosum (0.72 ± 0.04 × 10-3 mm2/s vs. 0.71 ± 0.03 × 10-3 mm2/s), and hippocampus (0.61 ± 0.03 × 10-3 mm2/s vs. 0.60 ± 0.04 × 10-3 mm2/s) showed no significant differences between the Tama and control rats.

  1. Similarly, I also have doubt regarding the shown differences in NAA, GPC+PCH, NAA+NAAG, but not in Ins, Glu+Gln.

Thank you for your considerate suggestions for and comments on our manuscript. All data is presented in this comment. It is no doubt that the NAA, GPC+PCH, NAA+NAAG values in Tama rat are higher than those of control rat, but not in Ins, Glu+Gln.

Line 149,

The concentrations and %SD of each metabolite in the Tama and control rats are shown in Figure 5. The values of NAA (6.97 ± 0.12 mM vs. 5.98 ± 0.25 mM, p < 0.01) and NAA + NAAG (8.22 ± 0.17 mM vs. 7.14 ± 0.35 mM, p < 0.01) of the Tama rats were higher than those of the control rats and showed significant differences. The GPC + PCh values of the Tama rats were lower than those of the control rats and showed a significant difference (1.04 ± 0.09 mM vs. 1.45 ± 0.04 mM, p < 0.001). The values of Ins (4.60 ± 0.78 mM vs. 5.39 ± 0.35 mM), NAAG (1.25 ± 0.05 mM vs. 1.16 ± 0.14 mM), Cr + PCr (5.91 ± 0.16 mM vs. 6.21 ± 0.31 mM), and Glu + Gln (12.14 ± 0.29 mM vs. 11.49 ± 0.89 mM) showed no significant differences between the Tama and control rats.

Reviewer 2 Report

The authors describe demonstration of dysmyelination in the Tama rats using MRI, comparing with PMD models as well as other PMD animal models. This provides a good information for researchers in this fields. There are only some minor points.

Points

1) Figures 1 and 2. How many independent experiments (independent animals) do the authors perform? The reviewer thinks that the images are the representative of the three animals.

2) Figures 3 to 5. The authors show the experimental numbers in the figure legends. For example, n=3... Also, an unpaired t-test is not suitable for multiple analyses.

3) Figure 6. How many independent experiments do the authors perform?

Author Response

The authors describe demonstration of dysmyelination in the Tama rats using MRI, comparing with PMD models as well as other PMD animal models. This provides a good information for researchers in this fields. There are only some minor points.

Thank you for your considerate suggestions for and comments on our manuscript. So, the following paragraph has been added to the manuscript.

Points

1) Figures 1 and 2. How many independent experiments (independent animals) do the authors perform? The reviewer thinks that the images are the representative of the three animals.

We used 4 Tama rats and 5 control rats written in the manuscript of Line 68. We showed Figure 1 as the representative of 9 rats for explaining how to set ROIs. We showed the representative images of one Tama rat and one control rat. A, B, and C are images of a Tama rat. And D, E, and F are those of a control rat. I think the ADC maps has image distortion cause of EPI read out. However, A, B and C images are obtained from same animal. Also, D, E and F images are obtained from same animal.

Line 68,

Four 3-week-old Tama rats with spontaneous mutations in the genes (weight = 54.8 ± 0.4 g) and five age-matched control rats (weight = 67.6 ± 4.3 g) were used.

2) Figures 3 to 5. The authors show the experimental numbers in the figure legends. For example, n=3... Also, an unpaired t-test is not suitable for multiple analyses.

We used 4 Tama rats and 5 control rats written in the manuscript of Line 68. We added the experimental numbers in the Figure description.

3) Figure 6. How many independent experiments do the authors perform?

We used 2 control rats and 2 Tama rats for LFB staining in the section 2.3.

We add more details as follows: Two slices of each rat were stained with LFB. And these are the representative of the Tama rat and the control rat.

Line 107,

We removed the brains of two Tama rats and two control rats after MRI measurements and perfused them with 4% phosphate-buffered saline. Axial slices (5–10-µm thick) were stained with LFB.
